# The Mortality Response to Absolute and Relative Temperature Extremes

**DOI:** 10.3390/ijerph16091493

**Published:** 2019-04-27

**Authors:** Scott C. Sheridan, Cameron C. Lee, Michael J. Allen

**Affiliations:** 1Department of Geography, Kent State University, Kent, OH 44242, USA; cclee@kent.edu; 2Department of Political Science and Geography, Old Dominion University, Norfolk, VA 23529, USA; mallen@odu.edu

**Keywords:** extreme temperature events, mortality, human biometeorology

## Abstract

While the impact of absolute extreme temperatures on human health has been amply studied, far less attention has been given to relative temperature extremes, that is, events that are highly unusual for the time of year but not necessarily extreme relative to a location’s overall climate. In this research, we use a recently defined extreme temperature event metric to define absolute extreme heat events (EHE) and extreme cold events (ECE) using absolute thresholds, and relative extreme heat events (REHE) and relative extreme cold events (RECE) using relative thresholds. All-cause mortality outcomes using a distributed lag nonlinear model are evaluated for the largest 51 metropolitan areas in the US for the period 1975–2010. Both the immediate impacts and the cumulative 20-day impacts are assessed for each of the extreme temperature event types. The 51 metropolitan areas were then grouped into 8 regions for meta-analysis. For heat events, the greatest mortality increases occur with a 0-day lag, with the subsequent days showing below-expected mortality (harvesting) that decreases the overall cumulative impact. For EHE, increases in mortality are still statistically significant when examined over 20 days. For REHE, it appears as though the day-0 increase in mortality is short-term displacement. For cold events, both relative and absolute, there is little mortality increase on day 0, but the impacts increase on subsequent days. Cumulative impacts are statistically significant at more than half of the stations for both ECE and RECE. The response to absolute ECE is strongest, but is also significant when using RECE across several southern locations, suggesting that there may be a lack of acclimatization, increasing mortality in relative cold events both early and late in winter.

## 1. Introduction

There has been ample research in recent decades that collectively shows a negative human health response to extreme temperature events (ETEs; e.g., [1,2,3]), with the broad shapes of the temperature–health relationship being rather similar across the globe once local climate is accounted for. In short, for extreme heat, there is evidence of an acute increase in mortality in most locations during and immediately after a heat event. Increases in human mortality are not solely due to direct hyperthermia, but originate from many other cardiovascular and respiratory-related causes as well [4]. Evidence exists that suggests that, for many extreme heat events, much of the short-term increase in mortality represents short-term displacement, meaning that following the heat event, mortality rates in many cases are below expected values for a period of time, diminishing the cumulative impact of heat [5,6]. In contrast, for extreme cold, the response in mortality is typically delayed, with a much longer lag in which increased mortality is observed, amplifying its impacts over time. Furthermore, research suggests that impacts of cold may be partitioned into more direct impacts at very cold temperatures and more indirect, lagged impacts due to increased spread of infections, at moderately cold temperatures [7]. 

One of the most critical issues with examining the impact of ETEs is that there is no formal definition of what an ETE is, and there are several papers that make comparisons of different ETE definitions their primary focus (e.g., [8,9]). Typically, any ETE metric incorporates some form of apparent temperature, which includes wind and humidity as well as temperature, to account properly for human thermoregulation [10], and these metrics, particularly when defined across broad climatically diverse areas, are usually defined in percentile terms *relative* to local climate, not *absolute* terms to account for differences in human response across climate zones [11].

Fewer studies have examined how definitions of ETEs should vary over time. It is well known that humans adapt to their local climate, and so some studies suggest that over many years, human health response will change as temperatures do [12]. Within a given year, there is some evidence that early-season ETEs are more hazardous than those later in the season (for heat, e.g., [1,13,14,15]), which may be due to a lack of acclimatization as well as the number of susceptible persons in a population.

To account for acclimatization, one should not only account for current thermal conditions, but also examine antecedent conditions, evaluating how relatively ‘oppressive’ current atmospheric conditions are. It is this concept that defines the Excess Heat Factor (EHF) developed by Nairn and Fawcett [16], which compares mean temperature over the most recent three days compared to the 30 before those. Warm conditions following relatively cool weather yield the highest EHF, and in several studies, EHF has been shown to be an effective predictor of temperature-related human mortality (e.g., [12,17,18]).

As defined by Nairn and Fawcett [16], the EHF still identifies what could be termed ‘absolute’ heat events, as it uses the 95th percentile of temperature based on year-round data; thus, it tends to identify events in the core of summer. It does not identify many early-season heat events that are extreme for the time of year, as many of those would not exceed the 95th percentile of the *annual* distribution. It would also omit all extreme temperature anomalies that fall outside of the core of summer, e.g., the March 2012 event across eastern North America that brought some of the most anomalous conditions seen in recent decades relative to the time of year, yet values that, when compared to an overall annual temperature curve, do not stand out as extreme (e.g., March 2012, [19]). It would also likely neglect events in relatively unique climates, such as the extreme warmth that sometimes affects southern California in the middle of winter due to downsloping easterly winds from the Sonoran Desert, which has been shown to increase human mortality [20,21,22].

Recent work by the authors included the development of a ‘relative’ extreme temperature metric to complement the absolute metric, in which the temperature thresholds are determined based on the time of year [23]. Excess Cold Factor (ECF) was defined analogous to the EHF but for low temperatures. Using these definitions, it is possible to identify extreme heat events (EHE), extreme cold events (ECE), relative extreme heat events (REHE), and relative extreme cold events (RECE) for North America. Using this climatology, in this research the human mortality responses to these events across the 51 largest metropolitan areas in the US for the period 1975–2010 is presented.

## 2. Data and Methods

### 2.1. Data and Calculation of Apparent Temperature

Hourly values of 2-m temperature, 2-m dew point, and 10-m wind speed (from u- and v-wind components) were obtained from the National Centers for Environmental Information NCEI) for the most representative airport for each of the 51 metropolitan areas (Table 1; Figure 1) for the period 1975–2010. To improve completeness, linear interpolation was performed to fill in single missing hourly values of each of the three variables.

Next, for each hour, an apparent temperature was then calculated based on the Steadman formula [10] for outdoor shade conditions, where
*AT* = −2.7 + 1.04*T* + 2.0*P* − 0.65*u*;(1)
in which *T* and *AT* are temperature and apparent temperature in °C, *P* is vapor pressure in kPa (calculated from dew point), and *u* is wind speed in m/s. The average of the 24-hourly values is the daily mean apparent temperature (*AT*), and is the basis for all calculations hereafter.

### 2.2. Calculation of Extreme Events

The extreme events in this paper were all first developed in [23], in turn based on the Excess Heat Factor (EHF) developed by Nairn and Fawcett [16]. In short, EHF is calculated as a product of the magnitude of the heat event, and an acclimatization term. The magnitude of the heat event, excess heat (*EH*) is calculated as
(2)$$EH=\max\left(0,\;\left(\sum_{i=-2}^{0}AT_{i}\right)/3-AT_{95}\right)$$
where *AT_i_* is the apparent temperature on day *i*, averaged over a three-day period, and *AT_95_* is the overall 95th percentile of apparent temperature for a particular location (based on the 1981–2010 normal period).

The acclimatization term is defined as:(3)$$EHaccl=\left(\sum_{i=-2}^{0}AT_{i}\right)/3-\left(\sum_{i=-32}^{-3}AT_{i}\right)/30,$$
the difference between the three-day mean apparent temperature and the 30 days prior.

*EHF* then is the product of these two terms,
*EHF* = max (0, *EH*) × max (1, *EH_accl_*),(4)
in units of K^2^. To define an extreme heat event (EHE), using the Nairn and Fawcett [16] definition, the *EHF* exceeds the 85th percentile of all positive *EHF* values for a location over the climatological period.

We extended this concept to create an Excess Cold Factor (ECF) and Extreme Cold Events (ECEs), defined similarly except with the 5th percentile of apparent temperature (*AT5*) as the basis for excess cold being identified, and the 15th percentile threshold of ECF used to identify ECE days.

To capture temperature events that are extreme relative to the time of year, we also assessed relative EHF (REHF) and ECF (RECF). Similar to the EHF and ECF above, these two variables are based on a seasonally moving threshold, calculated as the 92.5th (REHF)/7.5th (RECF) percentile over the climatological period for the 15 days centered on the day being evaluated. These different percentiles were used to provide roughly similar sample sizes to EHF and ECF. Relative Extreme Heat Events (REHE) were then identified as all days above the 85th percentile distribution of REHF, and relative extreme cold events (RECE) as days below the 15th percentile distribution of RECF.

### 2.3. Mortality Data

This research uses data on human mortality acquired from the National Center for Health Statistics for the United States for the period 1975–2010. Deaths were aggregated to all-cause, all-age totals at the metropolitan area level, using boundaries defined by the United States Census in 2010. A total of 28 days were removed from analysis based on excessive mortality totals for a metropolitan area that could clearly be attributed to something other than extreme temperature (see [12] for more information), accounting for transportation accidents, terrorism, tornado, fire, and hurricane. Due to irregular data, we excluded 1990 for Austin, Dallas, Houston, and San Antonio, and 2008 for Atlanta.

### 2.4. Calculating Relative Risks of Mortality

Relative risks (RR) of mortality were calculated for each of the 51 metropolitan areas separately for all-cause mortality over the full period of 1975–2010. Four different types of extreme temperature events (ETE) were individually considered as exposure:-Days that were classified as EHE-Days that were classified as ECE-Days that were classified as REHE that were not also EHE, and whose mean daily apparent temperature was at or above the mean annual apparent temperature for the location-Days that were classified as RECE that were not also ECE, and whose mean daily apparent temperature was at or below the mean annual apparent temperature for the location

The qualifiers that were placed on REHE and RECE were based on preliminary analysis, and intended to eliminate any confounding by including days that were categorized as absolute and relative extremes. The mean temperature threshold eliminated several mid-winter thaw periods that were identified as REHE, particularly across the southern US, to eliminate any potential confounding with overall winter mortality, although it ultimately did not impact results. The mean temperature threshold had had negligible impact on RECE.

For each of the above, we calculated relative risks using distributed-lag non-linear model (DLNM) that assesses the cumulative impact of weather events on mortality [24], using the *dlnm* package in R. The model is:Log (*Mortality*) = intercept + *ETE occurrence* + ns (*time*),
where:*mortality* is the daily all-cause mortality total for the metropolitan area, assuming a Poisson distribution of counts;ns (*time*) is a natural spline fit to the full 36-year period with 6 degrees of freedom per year (216 degrees of freedom in total), to account for long-term changes in baseline mortality as well as seasonal variations; and*ETE occurrence* refers to an array of binary variables created for each of the four ETE definitions above, where a day on which an ETE occurred is coded as 1, and a day in which an ETE did not occur is coded as 0. 

We set each model to examine the cumulative impact of heat over a 20-day period, as this allows a full assessment of potential mortality displacement with heat, and lagged impact of cold. This length is similar to lengths of lags used in other literature (e.g., [3,25]). The model that is fit to estimate the lagged effects of ETE is fixed with 3 knots with equally spaced log values over the 20 days. We performed sensitivity analyses on the number of degrees of freedom used in the spline that models the seasonality, as well as the number of knots to model the lagged effects, with negligible impact upon the statistical significance of the results detected (Table A1). Days with missing mortality data (the cases described in Section 2.3) and days with missing AT data (fewer than 1 percent of all days at all stations) were omitted from analysis.

Based on initial results, and geographic proximity, we pooled the results across the 51 metropolitan areas into 8 regions. We then performed a meta-analysis to estimate the pooled relative risk for each region using a random-effects model fitted through restricted maximum likelihood, using the *mvmeta* package in R.

## 3. Results

### 3.1. Climatology of ETEs

There is an average of 2.37 ECE days/year per station (Table 1 and Table 2), with the maximum at all metropolitan areas in December or January; together, these 2 months comprise 80% of all ECE days, with February representing a further 16%. The 5th percentile of apparent temperature used as a threshold for excess cold varies considerably across the varied climate of the US, ranging from −20.1 °C in Minneapolis to 14.6 °C in Miami.

On average, 2.44 EHE days/year occur across the metropolitan areas in this study, with a greater frequency across the southeastern and south-central US than elsewhere. The maximum frequency of EHE at all metropolitan areas is in July or August, except for Los Angeles, San Diego, and San Francisco, in which the maximum is September. Ninety-eight percent of all EHE days occur between June and September. The 95th percentile of apparent temperature varies from 16.7 °C in Seattle to 36.1 °C in Phoenix.

There are generally fewer days in the RECE sample size in this study, since many days identified as RECE are also ECE days. On average, around 1.12 days/year across the study area occur, with higher frequencies farther south. A bimodal pattern is observed across the year. The peak occurrence of RECE is in advance of the winter, in October and November, which together account for 46% of all RECE days. A secondary peak extends from March to April, in which an additional 27% of RECE days occur.

The sample size of REHE is also diminished compared with EHE, but to a lesser degree (than seen with RECE), since there is lesser coincidence of REHE and EHE on the same day. A study-wide mean of 1.68 days/year is observed, with generally higher values across warmer coastal regions than elsewhere. In advance of summer, March, April, and May together combine for 63% of all REHE days. The peak month gradually shifts northward as summer approaches, with a peak in February in Houston, March across much of the southern US, April across the northern US, and June in Seattle.

### 3.2. Associations between ETEs and Mortality

Extreme Cold Event (ECE) days are associated with a statistically significant increase in mortality across the US as a whole (Table 3 and Table 4), both on day 0 [RR: 1.029 (95% CI: 1.023, 1.035)] and in the 20-day cumulative period [RR: 1.201 (1.169, 1.233)], suggesting a greater than 20% increase in mortality risk. The lag-response structure shows an elevated risk on day 0 that increases and peaks 2–3 days after an ECE day, before retreating to being non-significant after around 15 days across the nation as a whole, and this pattern is broadly similar across each of the regions (Figure 2). The increase in mortality is most substantial across the southeastern US, particularly in the Southeast Coast region, where RR = 1.423 (1.348, 1.503) and the modeled lagged impact is significant out to 19 days. In separating impacts by month, the greatest 0-day impact is observed during the core ECE month of January [RR: 1.035 (1.029, 1.041)], although the greatest cumulative impact is observed at the start of winter, in December [RR: 1.351 (1.285, 1.419)] and decreasing through winter. This general seasonal pattern is observed across all regions except the Desert and Pacific, where the seasonal variations are much weaker, and is strongest within the Southeast and Southeast Coast regions. 

Extreme Heat Event (EHE) days are also associated with a statistically significant increase in mortality across the US (Figure 3), although the 20-day cumulative RR [1.091(1.066, 1.116)] is not substantially different from the 0-day impact alone [RR: 1.084 (1.071, 1.097)]. The impacts are greatest in the Northeast and Pacific, and lower across the southern US, in particular across the Southeast Coast region where the 20-day cumulative impact is not significant. Unlike with ECE, with EHE relative risks tend to align with the hottest period of the year, that is, they are greatest and most significant in July, and slightly less in June and August overall, although there are regional differences observed.

More nuanced results emerge from the relative events. For Relative Extreme Cold Events (RECE), overall there is a lack of immediate response, with only 3 metropolitan areas exhibiting a statistically significant increase in relative risk of mortality on Day 0, and a national RR of 1.009 (1.002, 1.015) that is statistically significant but not strong, with less than a 1% increase in risk. As with ECE, there is a sign of increased vulnerability with longer lags, as the 20-day cumulative lag at the national level is RR = 1.069 (1.022,1.117). The broad shape of the RR over time (Figure 4) is similar between RECE and ECE. Regionally, there is a substantially greater impact across the metropolitan areas of the Southeastern Coast [RR = 1.297 (1.210, 1.390)] than anywhere else. Temporally, this can be observed in both fall and spring across the southeast. In spring, statistically significant increases are seen regionally across the northeastern and midwestern US as well.

Relative Extreme Heat Events (REHE) show a statistically significant Day 0 response (RR = 1.018 (1.010, 1.025)], with the most substantive responses across the Pacific and Northeast, both of which were home to all statistically significant metropolitan areas. This Day 0 increase is nearly universally counteracted by a decrease in mortality on Days 1 and 2 (Figure 5), leaving no statistically significant increases in mortality in any metropolitan area or region when 20-day cumulative impacts are assessed.

## 4. Discussion

The results in this work regarding absolute extreme temperature events and human mortality, i.e., the extreme heat events (EHE) and extreme cold events (ECE), are similar to other research (e.g., [3,7]). This work broadly supports the different temporal attributes of health impact between heat and cold, with more immediate heat impacts and more delayed cold impacts, observed in most parts of the US (e.g., [3,10]). There is greater vulnerability to the heat in the northeastern, midwestern, and Pacific parts of the US, and decreased vulnerability farther south; this is generally the inverse of the impacts of cold. Regarding the heat, in this study the most substantial impacts occur in mid-summer. There is a slightly higher collective relative risk in June than in August, even though it is a cooler month overall, supporting other research regarding a greater vulnerability during early-season heat events (e.g., [10,13,26,27]), albeit weakly. We find particularly high levels of vulnerability in the southeastern US to cold events, similar to other research [15,21] using somewhat different approaches, supporting more broadly some other literature also showing cold impacts decreasing through the winter (e.g., [25]), as well as mortality spikes occurring with the passage of a mid-latitude cyclone that brings in extremely cold air [28].

Relative ETE, on the other hand, show some novel results with regard to the existing literature, as they have not been well studied. Relative Extreme Cold Events (RECE) broadly resemble the event-mortality relationship of ECE, suggesting that anomalously low temperatures throughout the year may have detrimental health impacts. This relationship is most substantial in the warmer regions of the country, particularly in the southeastern US (Table 5), where most metropolitan areas have an increased mortality risk from 20% to 50%, and is similar to what has been observed in other studies (e.g., [15]), although with events that include much less ‘wintry’ weather, for instance, in Miami, where mean daily apparent temperature is still near 15 °C. Lesser acclimatization to cold events has been shown in numerous studies, including some which have shown increased vulnerability (e.g., [29]), but few have explored the impact of what are overall relatively mild temperatures.

This work suggests that the observation of increased mortality during relative extreme heat events (REHE) is short-term displacement, as there are no statistically significant increases in mortality once lagged impacts are accounted for. This is different to some other research [30] that does show some mortality increase in very warm, but not hot, early-season days. However, this supports some general heat–health research (e.g., [6,31]), as well as some other explicit analyses of spring warm events, such as in England in April 2018 [32]. Indeed, in England, a greater absolute magnitude of decrease was found after the warm event than the increase during the event, similar to the observed protective effect of an REHE observed in this research, which materializes regardless of how the annual cycle is modeled (Table A1).

While there is a general expectation in the future of greater heat extremes and fewer cold extremes [33], over time there has been some research that has suggested that while vulnerability to heat has decreased, vulnerability to cold has not [34]. Furthermore, several other studies show greater overall vulnerability to cold than heat (e.g., [3]), though the magnitude depends on whether extreme events or all events are considered. Furthermore, as increased numbers of extreme events may outweigh the decreased risk to any single event [29], the results from this research suggest that more research into the seasonal variability in mortality beyond absolute extreme events is warranted. 

## 5. Conclusions

Using an extreme temperature event metric based on the Excess Heat Factor developed by Nairn and Fawcett [16], this research defined absolute extreme heat events (EHE) and extreme cold events (ECE) using thresholds that do not vary over the course of the year, and relative extreme heat events (REHE) and relative extreme cold events (RECE) whose thresholds change over the season cycle. For the 51 US metropolitan areas studied in this research, we showed that for heat events, there is a substantial increase in mortality at many locations with a 0-day lag, disappearing relatively quickly on subsequent days, which decreases the overall cumulative impact. For REHE at all metropolitan areas it appears that any day-0 increase in mortality is entirely offset by decreased mortality in the 20 subsequent days. Results for the cold events, both relative and absolute, show little if any increase in mortality on Day 0, with an increasing impact on mortality in subsequent days, resulting in statistically significant increases in 20-day cumulative mortality at more than half of the stations for both ECE and RECE. While ECE shows stronger results, particularly across the southern US there is a very substantial increase in mortality to RECE across several southern locations. These increases in relative cold events across the southern US are particularly noteworthy due to the limited research done on them to date, and warrant further consideration in terms of temporal changes over time, as well as potential intervention mechanisms.

## Figures and Tables

**Figure 1 ijerph-16-01493-f001:**
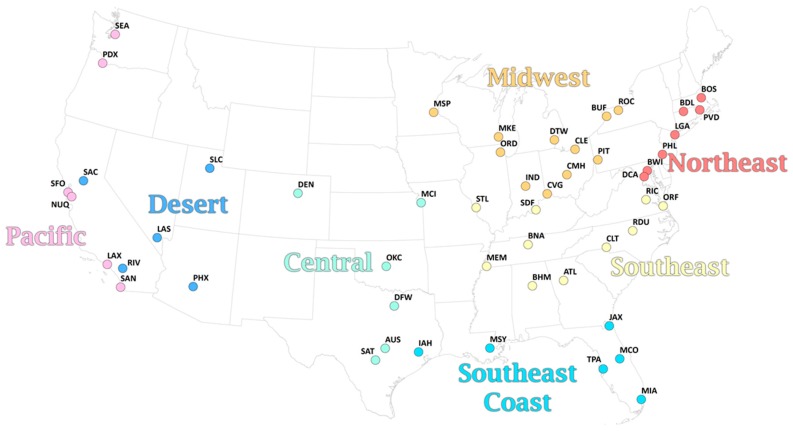
Map of the metropolitan areas in this study and their respective regional association.

**Figure 2 ijerph-16-01493-f002:**
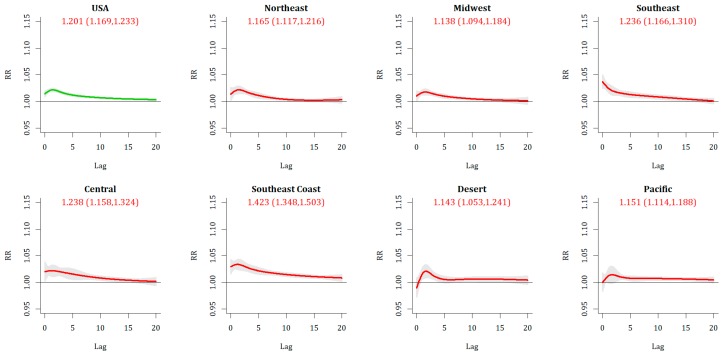
Relative risk (and confidence interval) associated with ECE day, for US as a whole (upper left) and each of the seven regions.

**Figure 3 ijerph-16-01493-f003:**
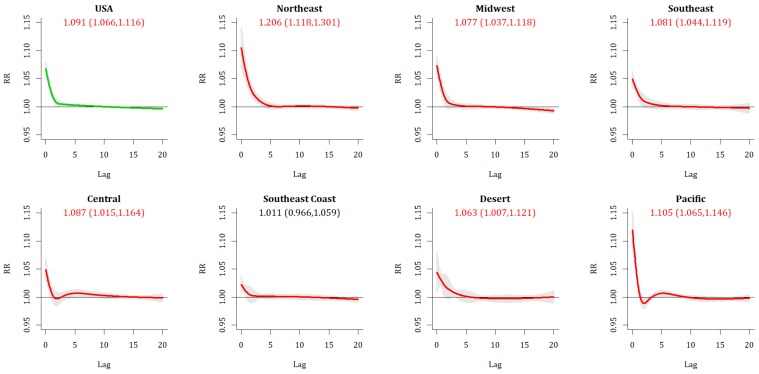
Same as Figure 2, except for EHE.

**Figure 4 ijerph-16-01493-f004:**
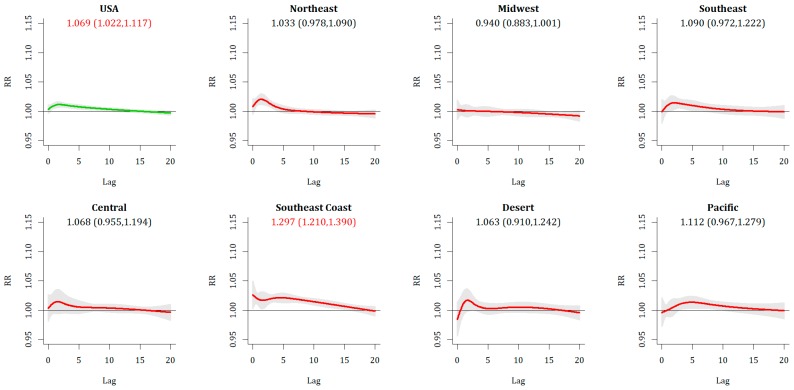
Same as Figure 2, except for RECE.

**Figure 5 ijerph-16-01493-f005:**
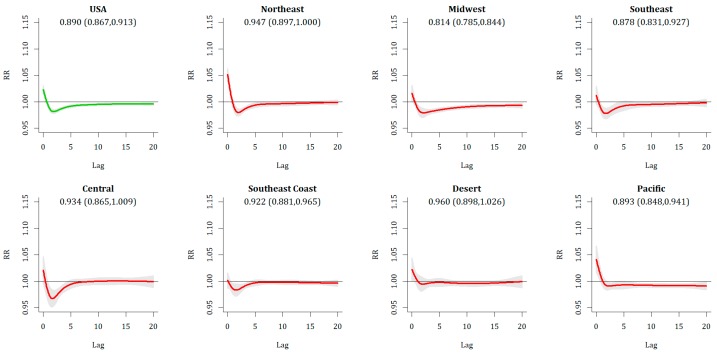
Same as Figure 2, except for REHE.

**Table 1 ijerph-16-01493-t001:** The 51 metropolitan areas used in this research, their 2010 populations, sample sizes of ETE event days, and apparent temperature percentiles (°C).

Metropolitan Area	Airport	Pop.	Sample Sizes				AT Percentiles	
	Code	(mln)	ECE	EHE	RECE	REHE	5th	95th
Atlanta	ATL	5.3	77	88	39	54	−2.8	28.9
Austin	AUS	1.7	78	87	84	44	1.9	31.5
Baltimore	BWI	2.7	81	81	29	67	−7.5	27.9
Birmingham	BHM	1.1	76	90	36	66	−1.7	29.8
Boston	BOS	4.5	77	76	36	56	−11.2	24.5
Buffalo	BUF	1.1	91	85	33	64	−14.5	23.3
Charlotte	CLT	2.2	85	81	39	66	−2.5	28.8
Chicago	ORD	9.5	97	78	30	70	−14.6	25.9
Cincinnati	CVG	2.1	100	87	33	65	−10.3	26.8
Cleveland	CLE	2.1	96	75	26	67	−12.9	24.9
Columbus	CMH	1.9	99	75	26	69	−11.2	26.5
Dallas	DFW	6.4	73	118	39	45	−1.6	31.6
Denver	DEN	2.5	51	86	55	54	−10.7	22.6
Detroit	DTW	4.3	87	78	34	66	−13.6	25.1
Hartford	BDL	1.2	83	72	38	63	−11.7	25.4
Houston	IAH	5.9	71	83	58	56	3.4	31.8
Indianapolis	IND	1.9	94	97	32	69	−12.0	26.8
Jacksonville	JAX	1.3	71	103	50	56	3.9	30.2
Kansas City	MCI	2.0	93	94	38	54	−12.2	28.1
Las Vegas	LAS	2.0	98	74	47	59	2.2	33.0
Los Angeles	LAX	12.8	96	98	42	78	9.0	21.8
Louisville	SDF	1.2	99	93	35	62	−7.9	29.0
Memphis	MEM	1.3	76	122	38	49	−3.6	31.4
Miami	MIA	5.6	86	141	43	95	14.6	31.4
Milwaukee	MKE	1.6	97	73	34	59	−15.5	24.3
Minneapolis	MSP	3.3	92	85	35	69	−20.1	24.9
Nashville	BNA	1.7	79	104	35	62	−5.2	29.3
New Orleans	MSY	1.2	71	113	48	71	3.7	31.5
New York	LGA	19.6	86	75	34	53	−9.4	26.9
Oklahoma City	OKC	1.3	79	104	31	46	−6.7	29.4
Orlando	MCO	2.1	73	82	49	66	8.4	30.3
Philadelphia	PHL	6.0	85	76	34	66	−8.6	27.6
Phoenix	PHX	4.2	105	82	61	58	8.0	36.1
Pittsburgh	PIT	2.4	84	85	32	69	−11.7	25.0
Portland	PDX	2.2	94	78	26	51	−1.8	21.2
Providence	PVD	1.6	84	79	37	51	−10.5	24.8
Raleigh	RDU	1.1	96	76	42	61	−3.3	28.8
Richmond	RIC	1.2	88	87	31	72	−4.9	28.8
Riverside	RIV	4.2	95	87	55	81	5.9	27.2
Rochester	ROC	1.1	89	89	35	71	−13.9	23.9
Sacramento	SAC	2.1	87	64	53	53	3.2	25.1
Saint Louis	STL	2.8	92	94	33	58	−10.2	29.7
Salt Lake City	SLC	1.1	95	75	26	54	−8.4	25.1
San Antonio	SAT	2.1	71	124	50	42	3.0	30.9
San Diego	SAN	3.1	92	113	51	74	10.3	23.3
San Francisco	SFO	4.3	88	75	51	40	5.1	16.7
San Jose	NUQ	1.8	57	66	60	35	6.3	21.1
Seattle	SEA	3.4	88	79	19	50	−1.3	19.0
Tampa	TPA	2.8	83	91	55	55	8.9	31.2
Virginia Beach	ORF	1.7	89	71	38	56	−3.9	28.8
Washington	DCA	5.6	82	88	37	63	−6.2	29.0

**Table 2 ijerph-16-01493-t002:** Mean number of ETE days/year by month, averaged across all 51 metropolitan areas for the period of study.

ETE Type	JAN	FEB	MAR	APR	MAY	JUN	JUL	AUG	SEP	OCT	NOV	DEC	ANN
ECE	1.17	0.38	0.05	0.00	0.00	0.00	0.00	0.00	0.00	0.00	0.05	0.72	2.37
EHE	0.00	0.00	0.00	0.01	0.02	0.42	1.23	0.62	0.12	0.02	0.00	0.00	2.44
RECE	0.01	0.07	0.17	0.13	0.05	0.02	0.00	0.00	0.06	0.22	0.30	0.09	1.12
REHE	0.15	0.14	0.41	0.41	0.23	0.04	0.00	0.00	0.00	0.05	0.08	0.16	1.68

**Table 3 ijerph-16-01493-t003:** Cumulative 20-day relative risks for ETEs by region, and number of statistically significant metropolitan areas.

**ECE**	**Overall**	**Dec**	**Jan**	**Feb**
NATIONAL	1.201 (1.169,1.233)	1.351 (1.285,1.419)	1.157 (1.124,1.190)	1.114 (1.064,1.167)
Northeast	1.165 (1.117,1.216)	1.455 (1.339,1.580)	1.129 (1.042,1.222)	1.096 (1.036,1.159)
Midwest	1.138 (1.094,1.184)	1.290 (1.187,1.402)	1.092 (1.057,1.127)	1.056 (0.965,1.156)
Southeast	1.236 (1.166,1.310)	1.513 (1.298,1.764)	1.176 (1.117,1.238)	1.027 (0.911,1.158)
Central	1.238 (1.158,1.324)	1.313 (1.202,1.434)	1.172 (1.076,1.275)	1.095 (0.973,1.232)
Southeast Coast	1.423 (1.348,1.503)	1.724 (1.494,1.990)	1.320 (1.253,1.390)	1.250 (0.971,1.609)
Desert	1.143 (1.053,1.241)	1.181 (1.102,1.265)	1.083 (0.955,1.228)	1.155 (0.853,1.565)
Pacific	1.151 (1.114,1.188)	1.162 (1.073,1.259)	1.207 (1.131,1.288)	1.184 (1.099,1.275)
S.S. Metros	42	37	25	6
**EHE**	**Overall**	**JUN**	**JUL**	**AUG**
NATIONAL	1.091 (1.066,1.116)	1.071 (1.027,1.117)	1.111 (1.077,1.147)	1.037 (1.005,1.071)
Northeast	1.206 (1.118,1.301)	1.234 (1.072,1.421)	1.231 (1.128,1.345)	1.076 (0.992,1.167)
Midwest	1.077 (1.037,1.118)	1.001 (0.937,1.069)	1.108 (1.052,1.168)	1.032 (0.956,1.115)
Southeast	1.081 (1.044,1.119)	1.053 (0.948,1.170)	1.124 (1.060,1.193)	0.991 (0.926,1.061)
Central	1.087 (1.015,1.164)	1.196 (1.023,1.398)	1.093 (1.013,1.179)	1.020 (0.947,1.098)
Southeast Coast	1.011 (0.966,1.059)	1.042 (0.975,1.113)	1.025 (0.898,1.171)	0.968 (0.865,1.083)
Desert	1.063 (1.007,1.121)	0.999 (0.751,1.331)	1.060 (0.994,1.130)	1.156 (0.998,1.338)
Pacific	1.105 (1.065,1.146)	1.103 (0.998,1.218)	1.116 (1.038,1.199)	1.141 (1.051,1.238)
S.S. Metros	21	5	20	5
**RECE**	**Overall**	**FALL**	**SPRING**	
NATIONAL	1.069 (1.022,1.117)	1.027 (0.972,1.085)	1.199 (1.130,1.272)	
Northeast	1.033 (0.978,1.090)	0.958 (0.880,1.044)	1.170 (1.061,1.290)	
Midwest	0.940 (0.883,1.001)	0.873 (0.809,0.942)	1.155 (1.012,1.319)	
Southeast	1.090 (0.972,1.222)	1.003 (0.893,1.126)	1.258 (1.062,1.489)	
Central	1.068 (0.955,1.194)	1.048 (0.913,1.201)	1.087 (0.892,1.325)	
Southeast Coast	1.297 (1.210,1.390)	1.309 (1.202,1.425)	1.315 (1.170,1.477)	
Desert	1.063 (0.910,1.242)	1.015 (0.883,1.167)	1.255 (0.801,1.964)	
Pacific	1.112 (0.967,1.279)	1.090 (0.948,1.253)	1.246 (0.986,1.573)	
S.S. Metros	12	7	13	
**REHE**	**Overall**	**SPRING**		
NATIONAL	0.890 (0.867,0.913)	0.910 (0.886,0.934)		
Northeast	0.947 (0.897,1.000)	0.980 (0.942,1.020)		
Midwest	0.814 (0.785,0.844)	0.830 (0.798,0.864)		
Southeast	0.878 (0.831,0.927)	0.913 (0.864,0.966)		
Central	0.934 (0.865,1.009)	0.992 (0.913,1.079)		
Southeast Coast	0.922 (0.881,0.965)	0.948 (0.908,0.991)		
Desert	0.960 (0.898,1.026)	0.931 (0.866,1.000)		
Pacific	0.893 (0.848,0.941)	0.881 (0.800,0.970)		
S.S. Metros	0	0		

**Table 4 ijerph-16-01493-t004:** Zero-day relative risks for ETEs by region, and number of statistically significant metropolitan areas.

**ECE**	**Overall**	**Dec**	**Jan**	**Feb**
NATIONAL	1.029 (1.023,1.035)	1.025 (1.015,1.035)	1.035 (1.029,1.041)	1.023 (1.013,1.033)
Northeast	1.031 (1.017,1.044)	1.038 (1.016,1.061)	1.029 (1.014,1.044)	1.029 (1.009,1.048)
Midwest	1.025 (1.016,1.034)	1.013 (0.993,1.034)	1.033 (1.022,1.044)	1.002 (0.981,1.024)
Southeast	1.051 (1.039,1.064)	1.044 (1.020,1.069)	1.055 (1.039,1.071)	1.032 (0.999,1.067)
Central	1.036 (1.020,1.053)	1.036 (1.009,1.064)	1.031 (1.005,1.058)	1.047 (1.011,1.084)
Southeast Coast	1.045 (1.032,1.058)	1.054 (1.032,1.076)	1.041 (1.023,1.059)	1.031 (0.986,1.078)
Desert	1.002 (0.984,1.021)	0.996 (0.974,1.019)	1.013 (0.972,1.056)	1.006 (0.967,1.046)
Pacific	1.010 (0.996,1.024)	1.002 (0.981,1.024)	1.021 (0.993,1.050)	1.023 (1.000,1.047)
S.S. Metros	23	9	14	4
**EHE**	**Overall**	**JUN**	**JUL**	**AUG**
NATIONAL	1.084 (1.071,1.097)	1.077 (1.057,1.097)	1.098 (1.084,1.113)	1.068 (1.048,1.088)
Northeast	1.142 (1.101,1.184)	1.130 (1.075,1.189)	1.147 (1.108,1.187)	1.132 (1.083,1.182)
Midwest	1.090 (1.067,1.112)	1.061 (1.032,1.090)	1.110 (1.083,1.139)	1.072 (1.037,1.108)
Southeast	1.073 (1.059,1.086)	1.021 (0.989,1.055)	1.108 (1.084,1.133)	1.025 (1.001,1.050)
Central	1.055 (1.032,1.079)	1.078 (1.027,1.132)	1.067 (1.040,1.095)	1.019 (0.994,1.045)
Southeast Coast	1.029 (1.017,1.040)	1.028 (1.004,1.053)	1.038 (1.014,1.062)	1.022 (0.997,1.048)
Desert	1.065 (1.040,1.091)	1.099 (1.040,1.160)	1.069 (1.047,1.093)	1.076 (1.019,1.136)
Pacific	1.123 (1.095,1.152)	1.161 (1.122,1.202)	1.113 (1.087,1.139)	1.142 (1.067,1.223)
S.S. Metros	43	19	42	21
**RECE**	**Overall**	**FALL**	**SPRING**	
NATIONAL	1.009 (1.002,1.015)	1.006 (0.998,1.014)	1.012 (0.998,1.026)	
Northeast	1.015 (1.002,1.029)	0.993 (0.975,1.011)	1.043 (1.023,1.064)	
Midwest	1.007 (0.992,1.021)	1.006 (0.989,1.024)	1.006 (0.973,1.040)	
Southeast	1.002 (0.983,1.021)	0.988 (0.956,1.021)	1.022 (0.981,1.064)	
Central	1.011 (0.989,1.034)	1.022 (0.994,1.050)	0.992 (0.950,1.035)	
Southeast Coast	1.026 (1.010,1.043)	1.035 (1.013,1.057)	1.016 (0.991,1.042)	
Desert	0.990 (0.960,1.020)	1.000 (0.967,1.035)	0.961 (0.926,0.998)	
Pacific	0.998 (0.980,1.017)	0.998 (0.978,1.019)	0.998 (0.962,1.035)	
S.S. Metros	3	3	6	
**REHE**	**Overall**	**SPRING**		
NATIONAL	1.018 (1.010,1.025)	1.020 (1.011,1.030)		
Northeast	1.046 (1.035,1.057)	1.050 (1.038,1.063)		
Midwest	1.015 (1.003,1.026)	1.018 (1.001,1.035)		
Southeast	1.003 (0.989,1.017)	1.002 (0.985,1.020)		
Central	1.009 (0.985,1.034)	0.999 (0.973,1.026)		
Southeast Coast	0.992 (0.979,1.005)	0.995 (0.979,1.010)		
Desert	1.019 (1.000,1.038)	1.023 (1.002,1.045)		
Pacific	1.045 (1.020,1.070)	1.052 (1.019,1.085)		
S.S. Metros	10	10		

**Table 5 ijerph-16-01493-t005:** Cumulative relative risks for metropolitan areas that have statistically significant increases in mortality during RECE.

Metro	Relative Risk (95% CI)
Atlanta	1.239 (1.028, 1.494)
Cleveland	1.233 (1.005, 1.513)
Dallas	1.213 (1.026, 1.435)
Jacksonville	1.505 (1.157, 1.958)
Los Angeles	1.399 (1.268, 1.543)
Memphis	1.313 (1.042, 1.653)
Miami	1.367 (1.207, 1.548)
Nashville	1.347 (1.036, 1.752)
New Orleans	1.352 (1.103, 1.658)
Orlando	1.307 (1.050, 1.628)
Sacramento	1.226 (1.040, 1.445)
Tampa	1.274 (1.115, 1.455)

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
