# Peer review of "The Mortality Response to Absolute and Relative Temperature Extremes"

_ijerph, 2019, doi:10.3390/ijerph16091493_

Round 1

Reviewer 1 Report

Although this research is very important to comprehend the levels of social resilience through extreme events, there some questions that I should emphasize (especially about the development of "data and methods" sector). Nevertheless, I really liked the geographical importance of this research, also and the importance of this research for US goverment agencies.

- page 02, lines (83-84): "... relative extreme heat events (REHE), and relative extreme cold 83 events (REHE) for North America". - Is there a minor mistake in REHE call? Check it, please.

- Table 1: please, develop better the table 1 design. It shows a bit desorganized.

- lines 151-155: please show the elements of Steadman equation (T, AT, u) didactically.

- lines 216-217: "Days with missing mortality data (the cases described in Section 2.1) and days with missing AT data (fewer than 1 percent of all days at all stations) were omitted from analysis". - what cases described in section 2.1?? Explain it, please.

- lines 218-220: "Based on initial results, and geographic proximity, we pooled the results across the 51 metropolitan areas into 8 regions, and performed a meta-analysis on each region using the mvmeta package in R". - How the authors have chosen these eight regions? How the authors have performed this "meta-analysis on each region using the mvmeta package in R"?? 

- I strongly suggest to develop a kind of panorama or a framework which could show all this method process. The text way to explain your methodology sometimes is a bit difficult to comprehend. Be kind to the reader.

- I beg your pardon, but I did not comprehend what is the meaning of "degrees of freedom" (lines 210 and 214). Explain it, please.

- I recommend a better explanation of DLNM equation variables. This part of the method sector is confused in my opinion. Also, this is very important because your main results are expressed in RR (relative risks of mortality).

- When I am seeing the graphics of RR, I am asking myself: what is a big, a common or a minor risk of mortality? Is the range between 0,99 to 1.30 relevant or not? If it is relevant, why? Where are the limits to understand what is serious or not? I am seeking for limits and scales to understand better the value of the results and to have the possibility to analyse these numbers geographically. Without a classification of RR limits, the numbers do not show all that they could do.

- Lines 298-299: "The results in this work regarding absolute extreme temperature events and human mortality, i.e., the extreme heat events (EHE) and extreme cold events (ECE), are similar to other research". - what research??

Author Response

Reviewer 1

Although this research is very important to comprehend the levels of social resilience through extreme events, there some questions that I should emphasize (especially about the development of "data and methods" sector). Nevertheless, I really liked the geographical importance of this research, also and the importance of this research for US goverment agencies.

R: Thank you for the compliments, and your helpful suggestions for our paper.

- page 02, lines (83-84): "... relative extreme heat events (REHE), and relative extreme cold 83 events (REHE) for North America". - Is there a minor mistake in REHE call? Check it, please.

R: This was in error – we have changed it to RECE.

- Table 1: please, develop better the table 1 design. It shows a bit desorganized.

R: We agree.  The manuscript we uploaded had the columns much more clearly aligned and spaced.  I assume in formatting once accepted this will be restored.  In the meantime, for this revision we did add tabs to make it easier to read, and better aligned.

- lines 151-155: please show the elements of Steadman equation (T, AT, u) didactically.

R: We added that T and AT are temperature and apparent temperature; and we have put the equation on its own line; beyond that we are not certain what you mean.

- lines 216-217: "Days with missing mortality data (the cases described in Section 2.1) and days with missing AT data (fewer than 1 percent of all days at all stations) were omitted from analysis". - what cases described in section 2.1?? Explain it, please.

R: Apologies – that should read section 2.3, not 2.1.  We have it explained on line 184-185.

- lines 218-220: "Based on initial results, and geographic proximity, we pooled the results across the 51 metropolitan areas into 8 regions, and performed a meta-analysis on each region using the mvmeta package in R". - How the authors have chosen these eight regions? How the authors have performed this "meta-analysis on each region using the mvmeta package in R"??

R: The regions were chosen based on similar regional climates, and initial RR observed at the metropolitan area level.  We apologize that we left out all details on mvmeta, we have added text.

- I strongly suggest to develop a kind of panorama or a framework which could show all this method process. The text way to explain your methodology sometimes is a bit difficult to comprehend. Be kind to the reader.

R: We hope that the improved description helps in the understanding of our work.  As the dlnm and mvmeta methods have been used in a number of papers to model the weather-health relationship without a figure, we did not feel it was necessary given space considerations in the text, although we are willing to add something if the editor requests it.

- I beg your pardon, but I did not comprehend what is the meaning of "degrees of freedom" (lines 210 and 214). Explain it, please.

R: The degrees of freedom refer to the number of degrees of freedom allowed in fitting a spline to the long-term data.  We have clarified this in the text by adding some more details throughout this section.

- I recommend a better explanation of DLNM equation variables. This part of the method sector is confused in my opinion. Also, this is very important because your main results are expressed in RR (relative risks of mortality).

R: We have added additional details to this section to make it clearer what we did.

- When I am seeing the graphics of RR, I am asking myself: what is a big, a common or a minor risk of mortality? Is the range between 0,99 to 1.30 relevant or not? If it is relevant, why? Where are the limits to understand what is serious or not? I am seeking for limits and scales to understand better the value of the results and to have the possibility to analyse these numbers geographically. Without a classification of RR limits, the numbers do not show all that they could do.

R: The relative risks are standard throughout the field and are usually presented as such.  In a few places, we have added some text to help translate the RR values into changes in risk.

- Lines 298-299: "The results in this work regarding absolute extreme temperature events and human mortality, i.e., the extreme heat events (EHE) and extreme cold events (ECE), are similar to other research". - what research??

R:  We intended that to be the lead-in to the text below; but in rereading it we agree it looks like an unsupported statement.  We have added some references.

Reviewer 2 Report

Wilson et al. (2013) Environmental Health 2013, 12:98

found that "all-cause mortality had similar magnitude associations
with single day and three day extreme and severe events (95th percentile lag0 (1.06: 1.03 – 1.09))".
Did you consider reviewing the models from this work? Perhaps you could refer to this work as well with regard to the lag linear models.

A very interesting paper.

Author Response

Reviewer 2

Wilson et al. (2013) Environmental Health 2013, 12:98

found that "all-cause mortality had similar magnitude associations
with single day and three day extreme and severe events (95th percentile lag0 (1.06: 1.03 – 1.09))".
Did you consider reviewing the models from this work? Perhaps you could refer to this work as well with regard to the lag linear models.

A very interesting paper.

R: Thank you for this reference; we were unaware of it.  We have cited it in the text as a reference for the use of EHF.

Round 2

Reviewer 1 Report

I have received the second version of the paper untitled "The mortality response to absolute and relative temperature extremes" and I am seeing that the authors improved this second version.

Moreover, it is undeniable the importance of researches like that which have a great importance to Federal agencies and strategical actions at the territory as well.

Congratulations!